# Discovery of a New Drug-like Series of OGT Inhibitors by Virtual Screening

**DOI:** 10.3390/molecules27061996

**Published:** 2022-03-19

**Authors:** Elena M. Loi, Tihomir Tomašič, Cyril Balsollier, Kevin van Eekelen, Matjaž Weiss, Martina Gobec, Matthew G. Alteen, David J. Vocadlo, Roland J. Pieters, Marko Anderluh

**Affiliations:** 1Department of Pharmaceutical Chemistry, Faculty of Pharmacy, University of Ljubljana, 1000 Ljubljana, Slovenia; elena.maria.loi@ffa.uni-lj.si (E.M.L.); tihomir.tomasic@ffa.uni-lj.si (T.T.); matjaz.weiss@ffa.uni-lj.si (M.W.); martina.gobec@ffa.uni-lj.si (M.G.); 2Department of Chemical Biology & Drug Discovery, Utrecht Institute for Pharmaceutical Sciences, Utrecht University, 3584 CG Utrecht, The Netherlands; c.balsollier@uu.nl (C.B.); k.vaneekelen@students.uu.nl (K.v.E.); r.j.pieters@uu.nl (R.J.P.); 3Department of Chemistry, Simon Fraser University, Burnaby, BC V5A 1S6, Canada; matthew.alteen@utoronto.ca (M.G.A.); dvocadlo@sfu.ca (D.J.V.)

**Keywords:** *O*-GlcNAc transferase, OGT inhibitors, virtual screening

## Abstract

*O*-GlcNAcylation is an essential post-translational modification installed by the enzyme *O*-β-*N*-acetyl-d-glucosaminyl transferase (OGT). Modulating this enzyme would be extremely valuable to better understand its role in the development of serious human pathologies, such as diabetes and cancer. However, the limited availability of potent and selective inhibitors hinders the validation of this potential therapeutic target. To explore new chemotypes that target the active site of OGT, we performed virtual screening of a large library of commercially available compounds with drug-like properties. We purchased samples of the most promising virtual hits and used enzyme assays to identify authentic leads. Structure-activity relationships of the best identified OGT inhibitor were explored by generating a small library of derivatives. Our best hit displays a novel uridine mimetic scaffold and inhibited the recombinant enzyme with an IC_50_ value of 7 µM. The current hit represents an excellent starting point for designing and developing a new set of OGT inhibitors that may prove useful for exploring the biology of OGT.

## 1. Introduction

The *O*-β-*N*-Acetyl-d-glucosaminyl transferase (OGT) is the only mammalian enzyme responsible for the transfer of N-acetylglucosamine from uridine diphosphate N-acetylglucosamine (UDP-GlcNAc) onto serine and threonine residues of nucleocytoplasmic proteins [1]. This post-translational modification (PTM) occurs on hundreds of cellular targets [2] and plays a crucial role in stress response, modulation of gene expression, signal transduction, and other essential cellular processes [3,4,5,6,7]. As a PTM, *O*-GlcNAcylation is similar to phosphorylation as it shares, in some cases, the very same protein substrates. Notably, cross-talk has been detected between the two modifications, suggesting *O*-GlcNAc may exert some of its effects by influencing protein phosphorylation [8,9]. Like phosphorylation, *O*-GlcNAcylation is a dynamic modification and is regulated by another enzyme, *O*-GlcNAcase (OGA), which is responsible for removing *O*-GlcNAc residues [10]. The activity of these enzymes is regulated in a manner that allows the cellular levels of *O*-GlcNAcylation to reflect the nutritional state of the cell. Indeed, when the intracellular concentration of UDP-GlcNAc increases due to elevated glucose levels, OGT expression and activity increase [11,12]. This makes OGT an emerging therapeutic target for conditions like diabetes [13], cancer [14,15], or heart failure [16], all of which are characterized by altered metabolism. Though it has been shown that OGT activity is altered in the in vitro and in vivo disease models [17,18,19], there remain many unanswered questions about the exact role that this enzyme plays in the associated pathogenic processes.

One of the main obstacles to validating OGT as a drug target is the lack of potent and specific OGT inhibitors. In the past decade, an increasing number of studies have focused on the discovery of such chemical tools, and the most prominent representatives are reported in Appendix A. For instance, precursor substrate analogs like 5SGlcNAc and 5SGlcNHex, or small molecules like OSMI-1 and OSMI-4, have been extremely useful in advancing the field [20,21,22,23,24]. Other examples include the covalent inhibitor BZX [25] or the bisubstrate inhibitor Goblin1 [26], which mimics the ternary Michaelis complex in which substrates are bound to OGT. However, since these compounds often lack cell permeability, display off-target effects, or are otherwise ill-suited for use in vivo, there is still a great need for novel OGT inhibitors that can be developed into broadly useful chemical biology tools. In our recent work, we used fragment-based drug design to produce the first OGT inhibitors with a 2-hydroxyquinoline-4-carboxamide scaffold, and we subsequently expanded this library by a fragment growing approach [27,28]. In this study, we present a new library of OGT inhibitors that targets the enzyme active site and displays various original uridine mimetic scaffolds. The library was designed by first conducting a large structure-based virtual screening campaign, followed by the synthesis of analogs (hit expansion) to explore the structure-activity relationships (SAR) for one of the most potent hits (Figure 1).

## 2. Results and Discussion

### 2.1. Virtual Screening

In designing our virtual screening campaign, we used an extensive library of different diversity sets consisting of more than two million compounds. All molecules were commercially available from different vendors and possessed drug-like properties, e.g., compliance with Lipinski and Veber criteria, and an absence of toxic compounds. For the docking experiment, we used the FRED tool from OEDOCKING (OpenEye Scientific Software, Santa Fe, NM, USA. http://www.eyesopen.com accessed on 14 March 2022) [29], which uses a fast and reliable structure-based docking algorithm and was proven to perform well in our comparative study of several docking tools on carbohydrate-binding protein DC-SIGN [30]. Analysis of the X-ray crystal structure of OGT bound to UDP-5SGlcNAc (PDB: 4GYY) revealed that the uracil moiety is anchored into the binding pocket by forming a bidentate hydrogen-bonding network with Ala896 (Figure 2d) [27,31]. Given the likely importance of this interaction coupled with its position deep within the active site, we elected to focus on the discovery of new uridine mimetic scaffolds that are able to mimic these interactions. Hydrogen bonds to Ala896 were therefore used as constraints in the virtual screening experiment. After the virtual screening was completed, the compounds were ranked based on their docking score, and the first 120 hits were selected for further investigation. In this way, we obtained a manageable number of compounds that allowed us to visually inspect their predicted binding poses. By overlapping the molecules with the co-crystallized UDP-5SGlcNAc ligand, the uridine mimetic moiety could be identified within each hit.

Following this analysis, we proceeded to cluster molecules displaying similar chemotypes and identified nine distinct families of compounds. Remarkably, one of the largest families, comprising 27 hits from among the top 120 compounds, exhibited a quinolone-4-carboxamide structure, which we already identified in our previous studies [26,27]. After carefully inspecting the predicted binding poses, we selected one to three hits from each cluster based on their synthetic accessibility and chemical diversity. Eighteen molecules were purchased from different vendors (Appendix A) and screened in vitro for OGT inhibition at 100 µM using a fluorescence-based transferase activity assay [32,33] (Table 1, Appendix A). Although six compounds were found to be active at this concentration, we identified Vs-5, Vs-51, and Vs-83 as the most potent and promising hits and therefore proceeded to measure their IC_50_ values. These three compounds showed comparable potencies. However, since Vs-51 was slightly more potent (IC_50_ = 68 µM, Table 1) and offered easier access to synthetic diversification than Vs-83 (IC_50_ = 88 µM, Table 1), we selected this compound for further optimization.

### 2.2. Hit Optimization

OpenEye docking tools FRED and Hybrid were used to guide us in designing a small new analog library based on the hit Vs-51. As shown in Figure 2d, another key interaction between OGT and UDP-5SGlcNAc is provided by the phosphate groups that form an extensive interaction network within the enzyme active site. According to the predicted binding mode of our virtual hit (Figure 2a,b), the triazole is responsible for anchoring the inhibitor into the same region by forming H-bonds with Lys842 and Thr922, the same residues that anchor biphosphate moiety of UDP-5S-GlcNAc.

The binding mode of Vs-51 in complex with OGT was also investigated by conducting a 100-ns molecular dynamics simulation (MD) starting from the docking complex (Figure 2a). Analysis of the ligand RMSD values (Appendix A) revealed two ligand binding modes. The docking binding mode (RMSD values below 2.5 Å) was maintained for the first 70 ns of the simulation. After this time point, the inhibitor lost both hydrogen bonds with the Ala896 side chain and moved toward the phosphate and GlcNAc binding pockets, where it formed an extensive network of hydrogen bonds and hydrophobic interactions (Figure 2c). The MD trajectory was also analyzed using the MD analysis tools implemented in LigandScout Expert 4.4.7. Appendix A shows the plot of the most frequent structure-based pharmacophore (SBPM) models derived from the MD simulation trajectory versus the number of occurrences. The most common model appears 137 times and has interactions consistent with those observed in the docking binding mode, namely two hydrogen bonds of the indolinone amide with Ala896 and hydrophobic interactions with Thr560, Phe694, Val895, Thr921 and Ala942 (Appendix A). The second most frequent SBPM, occurring 90 times, is a representative of the inhibitor binding mode in the last 30 ns of the simulation, in which the triazole ring formed two hydrogen bonds with the side chain Lys842, while the remaining part of the inhibitor formed hydrophobic contacts with Leu502, Thr560, Leu653, Tyr655, Phe694, Thr921, Thr922 and Ala942 (Appendix A).

We, therefore, decided to retain the important features of Vs-51 (indolinone and triazole moiety) and modify its substituents R_1_ and R_2_ (Table 2, Figure 1) to investigate the SAR of this region and improve upon the physicochemical properties of the initial hit. We also decided to explore the effect of installing chlorine at position 6 of the indolinone ring (R_3_), as we considered the possibility that it could either form a cation–dipole interaction with the side chain of Lys898 or have a positive steric effect by locking the molecule conformation to fit better the binding pocket.

We used a three-step synthetic route (Figure 1) to resynthesize Vs-51 and prepare the target analogs. To start, different isothiocyanates were refluxed with either 2-furoic or acetic acid hydrazide to obtain the intermediate thiocarbamides, which were then refluxed under basic conditions to form the triazole ring. Finally, the resulting triazoles were conjugated to the indolinone scaffold by a simple nucleophilic substitution using 5-(2-chloroacetyl)indolin-2-one. The resulting library consists of two main groups of compounds displaying either benzyl (**1**–**8**) or phenyl (**9**–**13**) substituents on position 4 of the triazole ring.

The newly synthesized library was evaluated against OGT by measuring IC_50_ values for each compound using the fluorescence-based transferase activity assay (Table 2). Interestingly, the resynthesized original hit (**1**) was 10-fold more potent than the commercial one (Vs-51), probably due to an inadequate purity of the latter, which we could not verify due to the small quantity of the stock sample that was used entirely for IC_50_ determination. With an IC_50_ value of 7 µM, this molecule is amongst the most promising OGT inhibitors reported to date, as it is only an order of magnitude weaker than OSMI-4a in the same assay [23].

Using derivatives **1**–**8,** we tested the effect of three different substituents in the *para* position of the phenyl ring. While the insertion of a methyl group led to a significant loss of inhibitory activity, methoxy and hydroxy substituents were better tolerated. The somewhat better potency of the phenolic derivative could be attributable to its potential to form a hydrogen bond with Leu653 (Figure 3a). However, its geometry is probably not optimal since this predicted additional contact in the active pocket does not seem to improve binding compared to the unsubstituted benzyl ring (**1**).

In order to obtain insights into the binding mode of the other promising hit, Vs-5, we designed a series of phenyl derivatives. These compounds present a phenyl ring on position 4 of the triazole, which we reasoned adds rigidity to the molecules and positions this group within a hydrophobic region of the binding site. Once again, the unsubstituted ring proved to be the best analog, as replacing the benzene with toluene, as in compound **10**, led to a decrease in inhibitory activity. According to our docking calculations, the orientation of the phenyl ring could also allow us to elongate the molecule toward a wider area of the active site delimited by His558 and Pro559 (Figure 3b). Hence, we introduced a morpholine ring, as seen in compound **13**, aiming at the same time to improve the water solubility of the inhibitor. However, this modification led to a complete loss of activity, probably due to deleterious steric effects that prevent the entire molecule from fitting into the enzyme active site. Regarding the modifications on position R_2_, our data suggest that the presence of a furan ring is beneficial for binding, as it probably provides a better fit into a small pocket close to the sugar-binding region. On the other hand, inserting a chlorine atom in position R_3_ (**2**, **5**, **8**, **12**) did not seem to significantly impact inhibitor activity, with the exception of compound **2,** in which it led to a substantial loss of potency. Hence, we can conclude that no interaction with Lys898 is gained by the incorporation of chlorine, and it does not lead to plausible binding conformation rigidization. Altogether, these data represent the foundation for the future optimization of these promising OGT inhibitors based on a novel uridine mimetic scaffold.

### 2.3. Cell-Based Assays

To assess whether **1** could inhibit OGT within the cellular environment, we selected two human cell lines: chronic myelogenous leukemia (K562) and human plasmacytoma (AMO1). The cells were treated with various concentrations of **1** (2–40 µM), then their metabolic activity was measured in a CellTiter 96 Aqueous One Solution Cell Proliferation (MTS) Assay (Figure 4a). Interestingly, in both cell lines, the OGT inhibitor induced a significant reduction in metabolic activity in a concentration-dependent fashion. However, Western blot analysis of the intracellular *O*-GlcNAcylation levels in AMO1 (Figure 4b,c) did not confirm significant inhibition of OGT in the same concentrations range. The absence of cellular activity against OGT indicates that compound **1** is probably highly protein bound, and consequently, not potent enough to be employed in cellular studies. This is consistent with the relatively high cLogP values of these compounds, including compound **1** (cLogP = 2.95) [34]. The reduced metabolic activity is probably due to off-target effects. Accordingly, we believe that it would be beneficial to further optimize the selectivity and potency of the hit compound.

## 3. Materials and Methods

### 3.1. Chemistry-General

All reagents and solvents were commercially available and used without further purification. Water used for isolations was purified. Column chromatography was carried out on silica gel 60 Merck 0.040–0.063 mm and preparative thin-layer chromatography (TLC) on silica gel plates F254 from Merck. ^1^H NMR and ^13^C NMR spectra were recorded using a Bruker Avance III 400 spectrometer (Bruker Corporation, Billerica, MA, USA), or an Agilent 400-MR spectrometer (Agilent Technologies, Inc., Santa Clara, CA, USA) operating at 400 MHz for ^1^H and 101 MHz for ^13^C, using TMS as the internal standard and DMSO-*d*_6_ as the solvent. Alternatively, they were recorded using a Bruker 600 Ultrashield spectrometer (Bruker Corporation, Billerica, MA, USA) operating at 600 MHz for ^1^H and 151 MHz for ^13^C. The chemical shifts (*δ* values) and coupling constants (*J* values) are given in ppm and hertz (Hz), respectively. HPLC analysis was performed on a Thermo Scientific Dionex UltiMate 3000 system (Thermo Fisher Scientific Inc., Waltham, MA, USA), using an Accucore C_18_ column (2.6 µm, 100 × 4.6 mm), at a flow rate of 0.8 mL/min, temperature 45 °C and an injection volume of 5 µL. Method: The eluent was a mixture of 0.1% TFA in water (A) and methanol (B). The gradient was 10% B to 90% B in 13 min, then 100% B for 2 min. The purity of all the tested compounds was established to be ≥95%, except for **11** (92%), and **12** (75%). High-resolution mass spectra were recorded with the Exactive^TM^ Plus Orbitrap mass spectrometer (Thermo Scientific, Waltham, MA, USA) and VG-Analytical Autospec Q spectrometer (VG Analytical Ltd, Manchester, UK).

#### 3.1.1. General Synthetic Procedures A, B and C

The general synthetic procedures A, B and C are shown in Figure 2.

##### General Procedure A for the Synthesis of Compounds **14**, **16**, **19**, **21**, **23**, **25**, **27**, **32**

A mixture of isothiocyanate and carbohydrazide was refluxed in ethanol for 3 h. The reaction mixture was then cooled down to 4 °C overnight to obtain the desired product as white crystals. The precipitate was filtered off, washed with cold ethanol, and dried under vacuum.

*N*-Benzyl-2-(furan-2-carbonyl)hydrazine-1-carbothioamide (**14**)

Benzyl isothiocyanate 236 mg (1.58 mmol); 2-furoic hydrazide 200 mg (1.58 mmol); Yield: 60%. ^1^H NMR (400 MHz, DMSO-*d*_6_) δ 10.32 (s, 1H), 9.43 (s, 1H), 8.68 (s, 1H), 7.89 (d, *J* = 0.9 Hz, 1H), 7.29 (d, *J* = 4.3 Hz, 4H), 7.25–7.18 (m, 2H), 6.66 (dd, *J* = 3.5, 1.7 Hz, 1H), 4.71 (d, *J* = 6.0 Hz, 2H).

2-Acetyl-N-benzylhydrazine-1-carbothioamide (**16**)

Benzyl isothiocyanate 604 mg (4.05 mmol); Acethydrazide 300 mg (4.05 mmol); Yield: 27%. ^1^H NMR (400 MHz, DMSO-*d*_6_) δ 9.73 (s, 1H), 9.27 (s, 1H), 8.48 (s, 1H), 7.34–7.25 (m, 4H), 7.24–7.18 (m, 1H), 4.71 (d, *J* = 6.0 Hz, 2H), 1.85 (s, 3H).

2-(Furan-2-carbonyl)-N-(4-methoxybenzyl)hydrazine-1-carbothioamide (**19**)

1-(Isothiocyanatomethyl)-4-methoxybenzene 400 mg (2.2 mmol); 2-furoic hydrazide 282 mg (2.2 mmol); Yield: 82%. ^1^H NMR (400 MHz, DMSO-*d*_6_) δ 10.29 (s, 1H), 9.37 (s, 1H), 8.61 (s, 1H), 7.89 (dd, *J* = 1.6, 0.7 Hz, 1H), 7.26–7.19 (m, 3H), 6.89–6.82 (m, 2H), 6.65 (dd, *J* = 3.5, 1.7 Hz, 1H), 4.63 (d, *J* = 5.9 Hz, 2H), 3.72 (s, 3H).

2-(Furan-2-carbonyl)-N-(4-methylbenzyl)hydrazine-1-carbothioamide (**21**)

1-(isothiocyanatomethyl)-4-methylbenzene 196 mg (1.2 mmol); 2-furoic hydrazide 150 mg (1.2 mmol); Yield: 60%. ^1^H NMR (400 MHz, DMSO-*d*_6_) δ 10.30 (s, 1H), 9.39 (s, 1H), 8.63 (s, 1H), 7.89 (dd, *J* = 1.6, 0.7 Hz, 1H), 7.22 (d, *J* = 3.4 Hz, 1H), 7.18 (d, *J* = 7.9 Hz, 2H), 7.09 (d, *J* = 7.9 Hz, 2H), 6.65 (dd, *J* = 3.5, 1.7 Hz, 1H), 4.65 (d, *J* = 6.0 Hz, 2H), 2.26 (s, 3H).

2-(Furan-2-carbonyl)-N-phenylhydrazine-1-carbothioamide (**23**)

Phenyl isothiocyanate 100 mg, 88 µL (0.74 mmol); 2-furoic hydrazide 150 mg (0.74 mmol); Yield: 78%. ^1^H NMR (400 MHz, DMSO-*d*_6_) δ 10.44 (s, 1H), 9.83 (s, 1H), 9.69 (s, 1H), 7.92 (dd, 1H), 7.44 (d, *J* = 7.1 Hz, 2H), 7.33 (t, *J* = 7.8 Hz, 2H), 7.25 (d, *J* = 3.3 Hz, 1H), 7.16 (t, *J* = 7.3 Hz, 1H), 6.68 (dd, *J* = 3.5, 1.7 Hz, 1H).

2-(Furan-2-carbonyl)-N-(p-tolyl)hydrazine-1-carbothioamide (**25**)

*p*-Tolyl isothiocyanate 100 mg (0.67 mmol); 2-furoic hydrazide 84 mg (0.67 mmol);

Yield: 90%. ^1^H NMR (400 MHz, DMSO-*d*_6_) δ 10.41 (s, 1H), 9.76 (s, 1H), 9.62 (s, 1H), 7.91 (dd, *J* = 1.6, 0.6 Hz, 1H), 7.34–7.26 (m, 2H), 7.24 (d, *J* = 3.3 Hz, 1H), 7.12 (d, *J* = 8.2 Hz, 2H), 6.67 (dd, *J* = 3.5, 1.7 Hz, 1H), 2.28 (s, 3H).

2-Acetyl-N-(p-tolyl)hydrazine-1-carbothioamide (**27**)

*p*-Tolyl isothiocyanate 290 mg (1.9 mmol); Acethydrazide 153 mg (2.1 mmol); Yield: 87%. ^1^H NMR (400 MHz, DMSO-*d*_6_) δ 9.83 (s, 1H), 9.52 (s, 1H), 9.43 (s, 1H), 7.28 (d, *J* = 8.0 Hz, 2H), 7.13 (d, *JJ* = 8.1 Hz, 2H), 2.28 (s, 3H), 1.88 (s, 3H).

2-(Furan-2-carbonyl)-N-(4-morpholinophenyl)hydrazine-1-carbothioamide (**32**)

4-(4-Isothiocyanatophenyl)morpholine 418 mg (1.9 mmol); 2-furoic hydrazide 248 mg (1.9 mmol); Yield: 75%. ^1^H NMR (400 MHz, DMSO-*d*_6_) δ 10.36 (s, 1H), 9.64 (s, 1H), 9.51 (s, 1H), 7.90 (dd, *JJ* = 1.8, 0.8 Hz, 1H), 7.27–7.23 (m, 2H), 7.23–7.20 (m, 1H), 6.91–6.86 (m, 2H), 6.66 (dd, *J* = 3.5, 1.7 Hz, 1H), 3.73 (m, 4H), 3.08 (m, 4H).

##### General Procedure B for the Synthesis of Compounds **15, 17, 20, 22, 24, 26, 28, 33**

A solution of the thiocarbamide obtained by general procedure A was refluxed in 2M NaOH (H_2_O: EtOH) for 3 h. The reaction mixture was then cooled to 0 °C and acidified with fuming HCl to obtain the desired product as white crystals. The precipitate was filtered off and dried under vacuum.

4-Benzyl-5-(furan-2-yl)-2,4-dihydro-3H-1,2,4-triazole-3-thione (**15**)

**14** 192 mg (0.7 mmol); Yield: 70%. ^1^H NMR (400 MHz, DMSO-*d*_6_) δ 14.22 (s, 1H), 7.91 (dd, *J* = 1.8, 0.7 Hz, 1H), 7.35–7.23 (m, 3H), 7.20–7.15 (m, 2H), 6.96 (dd, *J* = 3.5, 0.6 Hz, 1H), 6.65 (dd, *J* = 3.6, 1.8 Hz, 1H), 5.50 (s, 2H).

4-Benzyl-5-methyl-2,4-dihydro-3H-1,2,4-triazole-3-thione (**17**)

**16** 235 mg (1.05 mmol); Yield: 70%. ^1^H NMR (400 MHz, DMSO-*d*_6_) δ 13.62 (s, 1H), 7.41–7.24 (m, 5H), 5.23 (s, 2H), 2.17 (s, 3H).

5-(Furan-2-yl)-4-(4-methoxybenzyl)-2,4-dihydro-3H-1,2,4-triazole-3-thione (**20**)

**19** 540 mg (1.7 mmol); Yield: 60%. ^1^H NMR (400 MHz, DMSO-*d*_6_) δ 7.89 (d, *J* = 1.7 Hz, 1H), 7.12–7.07 (m, 2H), 6.95 (d, *J* = 3.5 Hz, 1H), 6.85–6.80 (m, 2H), 6.63 (dd, *J* = 3.5, 1.7 Hz, 1H), 5.38 (s, 2H), 3.66 (s, 3H).

5-(Furan-2-yl)-4-(4-methylbenzyl)-2,4-dihydro-3H-1,2,4-triazole-3-thione (**22**)

**21** 200 mg (0.7 mmol); Yield: 84%. ^1^H NMR (400 MHz, DMSO-*d*_6_) δ 14.19 (s, 1H), 7.91 (dd, *J* = 1.8, 0.7 Hz, 1H), 7.11 (d, *J* = 8.1 Hz, 2H), 7.06 (d, *J* = 8.2 Hz, 2H), 6.96 (dd, *J* = 3.6, 0.6 Hz, 1H), 6.66 (dd, *J* = 3.5, 1.8 Hz, 1H), 5.44 (s, 2H), 2.24 (s, 3H).

5-(Furan-2-yl)-4-phenyl-2,4-dihydro-3H-1,2,4-triazole-3-thione (**24**)

**23** 150 mg (0.6 mmol); Yield: 85%. ^1^H NMR (400 MHz, DMSO-*d*_6_) δ 14.18 (s, 1H), 7.81 (dd, *J* = 1.7, 0.6 Hz, 1H), 7.62–7.57 (m, 3H), 7.46–7.41 (m, 2H), 6.50 (dd, *J* = 3.6, 1.8 Hz, 1H), 5.87 (dd, *J* = 3.5, 0.6 Hz, 1H).

5-(Furan-2-yl)-4-(p-tolyl)-2,4-dihydro-3H-1,2,4-triazole-3-thione (**26**)

**25** 165 mg (0.6 mmol); Yield: 32%. ^1^H NMR (400 MHz, DMSO-*d*_6_) δ 14.13 (s, 1H), 7.82 (dd, *J* = 1.7, 0.6 Hz, 1H), 7.42–7.36 (m, 2H), 7.32–7.27 (m, 2H), 6.51 (dd, *JJ* = 3.6, 1.8 Hz, 1H), 5.88 (dd, *J* = 3.5, 0.6 Hz, 1H), 2.42 (s, 3H).

5-Methyl-4-(p-tolyl)-2,4-dihydro-3H-1,2,4-triazole-3-thione (**28**)

**27** 378 mg (1.7 mmol); Yield: 84%. ^1^H NMR (400 MHz, DMSO-*d*_6_) δ 7.38–7.34 (m, 2H), 7.32–7.26 (m, 2H), 2.39 (s, 3H), 2.08 (s, 3H).

5-(Furan-2-yl)-4-(4-morpholinophenyl)-2,4-dihydro-3H-1,2,4-triazole-3-thione (**33**)

**32** 444 mg (1.3 mmol); Yield: 87%. ^1^H NMR (400 MHz, DMSO-*d*_6_) δ 7.82 (d, *J* = 1.8 Hz, 1H), 7.25–7.19 (m, 2H), 7.11–7.04 (m, 2H), 6.51 (dd, *J* = 3.6, 1.8 Hz, 1H), 5.88 (d, *J* = 3.4 Hz, 1H), 3.76 (t, *J* = 4.8 Hz, 4H), 3.23 (t, *J* = 4.9 Hz, 4H).

##### General Procedure C for the Synthesis of Compounds **1, 2, 3, 4, 5, 7, 8, 9, 10, 11, 12, 13**

The 1,2,4-triazole-3-thione obtained by general procedure B was suspended in absolute ethanol in the presence of sodium bicarbonate or potassium carbonate. Moreso, 5-(2-chloroacetyl)indolin-2-one or 6-chloro-5-(2-chloroacetyl)indolin-2-one was added to the mixture, and the reaction was stirred for 48 to 72 h. The reaction progress was monitored by TLC and LC-MS. The formed precipitate was filtered off, washed with cold water, and purified by recrystallization or column chromatography.

5-(2-((4-Benzyl-5-(furan-2-yl)-4H-1,2,4-triazol-3-yl)thio)acetyl)indolin-2-one (**1**)

5-(2-Chloroacetyl)indolin-2-one 59 mg (0.28 mmol); **15** 60 mg (0.23 mmol); NaHCO_3_19 mg (0.23 mmol); Stirred at room temperature for 72 h. Purification: The light pink precipitate was filtered off and washed with cold ethanol and water. Yield: 56%. ^1^H NMR (400 MHz, DMSO-*d*_6_) δ 10.84 (s, 1H), 7.92 (dd, *J* = 8.3, 1.7 Hz, 1H), 7.89 (dd, *J* = 1.8, 0.7 Hz, 1H), 7.85 (s, 1H), 7.38–7.26 (m, 3H), 7.10 (d, *J* = 6.9 Hz, 2H), 6.95 (dd, *J* = 3.5, 0.7 Hz, 1H), 6.93 (d, *J* = 8.3 Hz, 1H), 6.66 (dd, *J* = 3.5, 1.8 Hz, 1H), 5.45 (s, 2H), 4.89 (s, 2H), 3.57 (s, 2H). ^13^C NMR (101 MHz, DMSO-*d*_6_) δ 191.51, 176.63, 150.67, 148.86, 147.24, 144.91, 140.98, 135.31, 129.64, 128.75, 128.61, 127.78, 126.28, 126.14, 124.51, 111.81, 111.62, 108.74, 47.51, 40.77, 35.37. HRMS (ESI+) *m*/*z* calcd for [C_23_H_18_N_4_O_3_S]: 431.11724 [M + H]^+^; found 431.11613.

5-(2-((4-Benzyl-5-(furan-2-yl)-4H-1,2,4-triazol-3-yl)thio)acetyl)-6-chloroindolin-2-one (**2**)

6-Chloro-5-(2-chloroacetyl)indolin-2-one 56 mg (0.23 mmol); **15** 60 mg (0.23 mmol); NaHCO_3_ 19 mg (0.23 mmol); Stirred at room temperature for 72h. Purification: The reaction mixture was concentrated in vacuo, and the residue was purified by flash chromatography. The sample was first eluted with a gradient of DCM/MeOH, and then the fractions containing the product were additionally purified by reverse phase chromatography (0.1% TFA/MeOH). Yield: 51%. ^1^H NMR (400 MHz, DMSO-*d*_6_) δ 10.83 (s, 1H), 7.89 (dd, *J* = 1.8, 0.7 Hz, 1H), 7.74 (s, 1H), 7.38–7.26 (m, 3H), 7.11–7.05 (m, 2H), 6.96 (dd, *J* = 3.5, 0.7 Hz, 1H), 6.91 (s, 1H), 6.66 (dd, *J* = 3.5, 1.8 Hz, 1H), 5.42 (s, 2H), 4.81 (s, 2H), 3.55 (s, 2H). ^13^C NMR (101 MHz, DMSO-*d*_6_) δ 193.65, 177.00, 151.06, 148.58, 147.83, 145.52, 141.50, 135.80, 131.64, 129.33, 128.70, 128.38, 127.03, 126.84, 125.51, 112.40, 112.24, 111.35, 48.07, 43.69, 35.63. HRMS (ESI+) *m*/*z* calcd for [C_23_H_17_ClN_4_O_3_S]: 465.07827 [M + H]^+^; found 465.07826.

5-(2-((4-Benzyl-5-methyl-4H-1,2,4-triazol-3-yl)thio)acetyl)indolin-2-one (**3**)

5-(2-Chloroacetyl)indolin-2-one 91 mg (0.4 mmol); **17** 74 mg (0.36 mmol); NaHCO_3_ 30 mg (0.7 mmol); Stirred at room temperature for 96h, then refluxed overnight. Purification: the reaction mixture was concentrated, and the residue resuspended in EtOAc and MeOH. The obtained white solid was filtered off and washed with cold water. Yield: 12%. ^1^1H NMR (400 MHz, DMSO-*d*_6_) δ 10.83 (s, 1H), 7.89 (dd, *J* = 8.3, 1.7 Hz, 1H), 7.82 (s, 1H), 7.41–7.28 (m, 3H), 7.14 (d, *J* = 7.0 Hz, 2H), 6.92 (d, *J* = 8.2 Hz, 1H), 5.20 (s, 2H), 4.77 (s, 2H), 3.57 (s, 2H), 2.28 (s, 3H). ^13^C NMR (101 MHz, DMSO-*d*_6_) δ 192.25, 177.22, 156.39, 153.18, 149.37, 135.98, 130.18, 129.36, 129.20, 128.37, 127.24, 126.70, 125.07, 109.29, 46.97, 41.24, 35.94, 11.17. HRMS (ESI+) *m*/*z* calcd for [C_20_H_18_N_4_O_2_S]: 379.12232 [M + H]^+^; found 379.12093.

5-(2-((5-(Furan-2-yl)-4-(4-methoxybenzyl)-4H-1,2,4-triazol-3-yl)thio)acetyl)indolin-2-one (**4**)

5-(2-Chloroacetyl)indolin-2-one 80 mg (0.38 mmol); **20** 100 mg (0.35 mmol); K_2_CO_3_ 73 mg (0.5 mmol); Stirred at room temperature for 72 h. Purification: The light pink precipitate was filtered off and washed with cold ethanol and water. Yield: 90%. ^1^H NMR (400 MHz, DMSO-*d*_6_) δ 10.82 (s, 1H), 7.91 (dd, *J* = 4.6, 3.4 Hz, 2H), 7.84 (s, 1H), 7.04 (d, *J* = 8.7 Hz, 2H), 6.96 (d, *J* = 3.4 Hz, 1H), 6.93 (d, *J* = 8.2 Hz, 1H), 6.88 (d, *J* = 8.7 Hz, 2H), 6.67 (dd, *J* = 3.4, 1.8 Hz, 1H), 5.36 (s, 2H), 4.87 (s, 2H), 3.70 (s, 3H), 3.56 (s, 2H). ^13^C NMR (101 MHz, DMSO-*d*_6_) δ 192.05, 177.17, 159.27, 151.07, 149.37, 147.66, 145.41, 141.56, 130.16, 129.16, 128.39, 127.64, 126.66, 125.03, 114.63, 112.36, 112.19, 109.28, 55.50, 47.58, 41.26, 35.90. HRMS (ESI+) *m*/*z* calcd for [C_24_H_20_N_4_O_4_S]: 461.12780 [M + H]^+^; found 461.12773.

6-Chloro-5-(2-((5-(furan-2-yl)-4-(4-methoxybenzyl)-4H-1,2,4-triazol-3-yl)thio)acetyl)indolin-2-one (**5**)

6-Chloro-5-(2-chloroacetyl)indolin-2-one 41 mg (0.17 mmol); **20** 40 mg (0.14 mmol); NaHCO_3_ 12 mg (0.14 mmol); Stirred at room temperature for 72 h. Purification: the reaction mixture was concentrated, and the residue was purified by column chromatography eluting with DCM/MeOH (gradient 2–10%). Yield: 26%. ^1^H NMR (400 MHz, DMSO-*d*_6_) δ 10.81 (s, 1H), 7.91 (dd, *J* = 1.8, 0.7 Hz, 1H), 7.72 (s, 1H), 7.05–6.99 (m, 2H), 6.97 (dd, *J* = 3.5, 0.7 Hz, 1H), 6.91 (s, 1H), 6.90–6.85 (m, 2H), 6.67 (dd, *J* = 3.5, 1.8 Hz, 1H), 5.32 (s, 2H), 4.79 (s, 2H), 3.70 (s, 3H), 3.54 (s, 2H). ^13^C NMR (101 MHz, DMSO-*d*_6_) δ 193.62, 176.93, 159.28, 150.86, 148.51, 147.67, 145.43, 141.51, 131.58, 128.68, 128.37, 127.56, 126.95, 125.45, 114.62, 112.36, 112.21, 111.29, 55.50, 47.56, 43.62, 35.58. HRMS (ESI+) *m*/*z* calcd for [C_24_H_19_ClN_4_O_4_S]: 495.08883 [M + H] ^+^; found 495.08897.

5-(2-((5-(Furan-2-yl)-4-(4-methylbenzyl)-4H-1,2,4-triazol-3-yl)thio)acetyl)indolin-2-one (**7**)

5-(2-Chloroacetyl)indolin-2-one 46 mg (0.22 mmol); **22** 60 mg (0.22 mmol); NaHCO_3_ 19 mg (0.22 mmol); Stirred at room temperature for 72 h. Purification: The light pink precipitate was filtered off and washed with cold ethanol and water. Yield: 21%. ^1^H NMR (400 MHz, DMSO-*d*_6_) δ 10.83 (s, 1H), 7.90 (d, *J* = 1.2 Hz, 1H), 7.96–7.82 (m, 2H), 7.14 (d, *J* = 7.9 Hz, 2H), 6.98 (d, *J* = 8.0 Hz, 2H), 6.94 (dd, *J* = 7.6, 6.1 Hz, 2H), 6.67 (dd, *J* = 3.5, 1.8 Hz, 1H), 5.39 (s, 2H), 4.88 (s, 2H), 3.57 (s, 2H), 2.25 (s, 3H). ^13^C NMR (101 MHz, DMSO-*d*_6_) δ 192.11, 177.21, 151.18, 149.42, 147.79, 145.47, 141.59, 137.65, 132.86, 130.21, 129.86, 129.20, 126.84, 126.72, 125.08, 112.39, 112.19, 109.31, 47.89, 41.34, 35.94, 21.08. HRMS (ESI+) *m*/*z* calcd for [C_24_H_20_N_4_O_3_S]: 445.13289 [M + H]^+^; found 445.13160.

6-Chloro-5-(2-((5-(furan-2-yl)-4-(4-methylbenzyl)-4H-1,2,4-triazol-3-yl)thio)acetyl)indolin-2-one (**8**)

6-Chloro-5-(2-chloroacetyl)indolin-2-one 54 mg (0.22 mmol); **22** 60 mg (0.22 mmol); NaHCO_3_ 19 mg (0.22 mmol); Stirred at room temperature for 72 h. Purification: The dark pink precipitate was filtered off and washed with cold ethanol and water. The precipitate was additionally purified by column chromatography, eluting with EtOAc/MeOH (gradient 0–5%). Yield: 9%. ^1^H NMR (400 MHz, DMSO-*d*_6_) δ 10.83 (s, 1H), 7.90 (d, *J* = 1.2 Hz, 1H), 7.73 (s, 1H), 7.13 (d, *J* = 7.9 Hz, 2H), 6.96 (s, 1H), 6.96 (d, *J* = 7.5 Hz, 2H), 6.91 (s, 1H), 6.67 (dd, *J* = 3.5, 1.8 Hz, 1H), 5.36 (s, 2H), 4.80 (s, 2H), 3.55 (s, 2H), 2.25 (s, 3H). ^13^C NMR (101 MHz, DMSO-*d*_6_) δ 193.66, 176.99, 150.98, 148.57, 147.79, 145.49, 141.54, 137.68, 132.78, 131.64, 129.85, 128.70, 127.01, 126.82, 125.50, 112.40, 112.21, 111.33, 47.87, 43.68, 35.62, 21.08. HRMS (ESI+) *m*/*z* calcd for [C_24_H_19_ClN_4_O_3_S]: 479.09392 [M + H]^+^; found 479.09282.

5-(2-((5-(Furan-2-yl)-4-phenyl-4H-1,2,4-triazol-3-yl)thio)acetyl)indolin-2-one (**9**)

5-(2-Chloroacetyl)indolin-2-one 52 mg (0.25 mmol); **24** 50 mg (0.21 mmol); K_2_CO_3_ 58 mg (0.42 mmol); Stirred for 48 h at room temperature. Purification: The reaction mixture was concentrated in vacuo, and the residue was purified by flash chromatography. The sample was first eluted with a gradient of DCM/MeOH, and then the fractions containing the product were additionally purified by reverse phase chromatography (0.1% TFA/MeOH). Yield: 28%. ^1^H NMR (400 MHz, DMSO-*d*_6_) δ 10.84 (s, 1H), 7.93 (dd, *J* = 8.2, 1.8 Hz, 1H), 7.86 (s, 1H), 7.76 (dd, *J* = 1.8, 0.7 Hz, 1H), 7.67–7.60 (m, 3H), 7.54–7.48 (m, 2H), 6.94 (d, *J* = 8.2 Hz, 1H), 6.52 (dd, *J* = 3.5, 1.8 Hz, 1H), 6.13 (dd, *J* = 3.5, 0.7 Hz, 1H), 4.88 (s, 2H), 3.58 (s, 2H). ^13^C NMR (101 MHz, DMSO-*d*_6_) δ 192.01, 177.23, 151.68, 149.44, 147.70, 145.31, 141.37, 133.77, 131.03, 130.53, 130.22, 129.18, 128.12, 126.75, 125.09, 112.08, 111.67, 109.34, 40.77, 35.96. HRMS (ESI+) *m*/*z* calcd for [C_22_H_16_N_4_O_3_S]: 417.10159 [M + H]^+^; found 417.10151.

5-(2-((5-(Furan-2-yl)-4-(p-tolyl)-4H-1,2,4-triazol-3-yl)thio)acetyl)indolin-2-one (**10**)

5-(2-Chloroacetyl)indolin-2-one 50 mg (0.24 mmol); **26** 50 mg (0.2 mmol); K_2_CO_3_ 55 mg (0.4 mmol); Stirred for 48 h at room temperature. Purification: The reaction mixture was concentrated in vacuo, and the residue was purified by flash chromatography eluting with a gradient of DCM/MeOH. Yield: 37%. ^1^H NMR (400 MHz, DMSO-*d*_6_) δ 10.84 (s, 1H), 7.92 (dd, *J* = 8.2, 1.7 Hz, 1H), 7.86 (s, 1H), 7.78 (dd, *J* = 1.8, 0.7 Hz, 1H), 7.44–7.41 (m, 2H), 7.39–7.34 (m, 2H), 6.94 (d, *J* = 8.2 Hz, 1H), 6.52 (dd, *J* = 3.5, 1.8 Hz, 1H), 6.12 (dd, *J* = 3.5, 0.7 Hz, 1H), 4.87 (s, 2H), 3.58 (s, 2H), 2.43 (s, 3H). ^13^C NMR (101 MHz, DMSO-*d*_6_) δ 192.04, 177.23, 155.66, 151.82, 145.28, 141.41, 140.84, 136.90, 131.15, 130.98, 130.22, 129.18, 127.83, 126.74, 125.09, 112.07, 111.62, 109.33, 40.65, 35.96, 21.31. HRMS (ESI+) *m*/*z* calcd for [C_23_H_18_N_4_O_3_S]: 431.11724 [M + H]^+^; found 431.11728.

5-(2-((5-Methyl-4-(p-tolyl)-4H-1,2,4-triazol-3-yl)thio)acetyl)indolin-2-one (**11**)

5-(2-Chloroacetyl)indolin-2-one 127 mg (0.6 mmol); **28** 95 mg (0.5 mmol); NaHCO_3_ 41 mg (0.5 mmol); Stirred at 40 °C for 48 h. The mixture was concentrated, and ice was added to the mixture and stirred until it was completely melted. The formed precipitate was filtered off and dried. The precipitate was purified by column chromatography (1:30 MeOH: DCM). The product was concentrated and dried. This resulted in a pale pink solid (yield: 79%). ^1^H NMR (400 MHz, DMSO-*d*_6_) δ 10.81 (s, 1H), 7.89 (dd, *J* = 8.2, 1.9 Hz, 1H), 7.83 (s, 1H), 7.44–7.36 (m, 2H), 7.36–7.28 (m, 2H), 6.92 (dd, *J* = 8.2, 3.5 Hz, 1H), 4.76 (s, 2H), 3.57 (s, 2H), 2.40 (s, 3H), 2.18 (s, 3H). ^13^C NMR (101 MHz, DMSO-*d*_6_) δ 192.09, 177.16, 152.80, 149.63, 149.31, 140.06, 130.99, 130.79, 130.11, 129.18, 127.22, 126.65, 125.01, 109.25, 40.43, 35.91, 21.17, 11.26. HRMS (ESI+) *m*/*z* calcd for [C_20_H_18_N_4_O_2_S]: 379.12232 [M + H]^+^; found 379.12228.

6-Chloro-5-(2-((5-methyl-4-(p-tolyl)-4H-1,2,4-triazol-3-yl)thio)acetyl)indolin-2-one (**12**)

6-Chloro-5-(2-chloroacetyl)indolin-2-one 138 mg (0.6 mmol); **28** 95 mg (0.5 mmol); NaHCO_3_ 41 mg (0.5 mmol); Stirred at 40 °C for 48 h. The mixture was concentrated, and ice was added to the mixture and stirred until it was completely melted. The formed precipitate was filtered off and dried. The precipitate was purified by column chromatography (1:30 MeOH: DCM). The product was concentrated and dried. This resulted in a pink solid (yield: 63%). ^1^H NMR (600 MHz, DMSO-*d*_6_) δ 10.83 (s, 1H), 7.72 (s, 1H), 7.40 (d, *J* = 8.0 Hz, 2H), 7.31 (d, *J* = 7.9 Hz, 2H), 6.90 (s, 1H), 4.70 (s, 2H), 3.56 (s, 2H), 2.40 (d, *J* = 5.5 Hz, 3H), 2.18 (s, 3H). ^13^C NMR (101 MHz, DMSO-*d*_6_) δ 193.67, 176.95, 152.83, 149.49, 148.48, 140.10, 131.52, 130.90, 130.81, 128.80, 127.16, 126.97, 125.48, 111.26, 42.94, 35.61, 21.17, 11.23. HRMS (ESI+) *m*/*z* calcd for [C_20_H_17_ClN_4_O_2_S]: 413.08335 [M + H]^+^; found 413.08328.

5-(2-((5-(Furan-2-yl)-4-(4-morpholinophenyl)-4H-1,2,4-triazol-3-yl)thio)acetyl)indolin-2-one (**13**)

5-(2-Chloroacetyl)indolin-2-one 40 mg (0.2 mmol); **33** 50 mg (0.15 mmol); NaHCO_3_ 13 mg (0.15 mmol); Stirred at 40° C for 48 h. The mixture was concentrated, and ice was added to the mixture and stirred until it was completely melted. The formed precipitate was filtered off and dried. The precipitate was purified by column chromatography (1:30 MeOH: DCM). The product was concentrated and dried. This resulted in a pale pink solid (yield: 35%). ^1^H NMR (600 MHz, DMSO-*d*_6_) δ 10.83 (s, 1H), 7.92 (dd, *J* = 8.2, 1.7 Hz, 1H), 7.85 (s, 1H), 7.78 (dd, *J* = 1.8, 0.8 Hz, 1H), 7.32–7.26 (m, 2H), 7.12–7.07 (m, 2H), 6.93 (d, *J* = 8.2 Hz, 1H), 6.52 (dd, *J* = 3.5, 1.8 Hz, 1H), 6.09 (dd, *J* = 3.5, 0.8 Hz, 1H), 4.85 (s, 2H), 3.78–3.74 (m, 4H), 3.58 (s, 2H), 3.26–3.22 (m, 4H). ^13^C NMR (151 MHz, DMSO-*d*_6_) δ 192.13, 177.23, 152.35, 152.25, 149.43, 148.07, 145.16, 141.58, 130.22, 129.21, 128.63, 126.74, 125.10, 123.83, 115.42, 112.07, 111.43, 109.34, 66.44, 47.84, 40.60, 35.97. HRMS (ESI+) *m*/*z* calcd for [C_26_H_23_N_5_O_4_S]: 502.15435 [M + H]^+^; found 502.15466.

#### 3.1.2. Synthetic Procedure for Compound **18**

The synthetic procedure for compound **18** is shown in Figure 3.

##### 1-(Isothiocyanatomethyl)-4-methoxybenzene (**18**)

An amount of 476 µL (3.6 mmol) of (4-methoxyphenyl)methanamine and 1.5 mL (10.8 mmol) of triethylamine were dissolved in 8 mL of dry THF at 0 °C under argon atmosphere. An amount of 217 µL (3.6 mmol) of carbon disulfide was slowly added to the cold mixture, then the reaction was stirred for one hour at room temperature. When the formation of the dithiocarbamate salt was completed (monitored by TLC and LC-MS), the mixture was once again cooled down to 0 °C before adding 755 mg (3.9 mmol) of tosyl chloride. The reaction was stirred for 30 min at room temperature, then the mixture was poured into a 1 M HCl solution and extracted 3 times with DCM. The organic layers were combined, dried, and concentrated, then the residue was purified by column chromatography, eluting with 100% hexane. The product was concentrated to yield a pale-yellow solid (yield: 90%). ^1^H NMR (400 MHz, CDCl_3_) δ 7.26–7.21 (m, 2H), 6.94–6.87 (m, 2H), 4.64 (s, 2H), 3.82 (s, 3H).

#### 3.1.3. Synthetic Procedure for Compound **6**


The synthetic procedure for compound **6** is depicted in Figure 4.

##### 5-(2-((5-(Furan-2-yl)-4-(4-hydroxybenzyl)-4H-1,2,4-triazol-3-yl)thio)acetyl)indolin-2-one (**6**)

An amount of 100 mg (0.2 mmol) of **4** was dissolved in dry DCM under argon atmosphere, and the solution was cooled to -20 °C. An amount of 660 µL of 1 M BBr_3_ was added to the mixture, and the reaction was stirred for 2 h at room temperature. ^1^H NMR (400 MHz, DMSO-*d*_6_) δ 10.79 (s, 1H), 7.90–7.85 (m, 2H), 7.80 (s, 1H), 6.93 (dd, *J* = 3.5, 0.6 Hz, 1H), 6.92–6.87 (m, 3H), 6.69–6.65 (m, 2H), 6.65–6.63 (m, 1H), 5.27 (s, 2H), 4.83 (s, 2H), 3.53 (s, 2H). ^13^C NMR (101 MHz, DMSO-*d*_6_) δ 192.05, 177.15, 157.51, 151.10, 149.37, 147.59, 145.45, 141.48, 130.15, 129.15, 128.48, 126.67, 125.83, 125.03, 115.94, 112.37, 112.31, 109.28, 47.76, 41.28, 35.91. HRMS (ESI+) *m*/*z* calcd for [C_23_H_18_N_4_O_4_S]: 447.11215 [M + H]^+^; found 447.11217.

#### 3.1.4. Synthetic Procedure for Compounds **29**, **30**, **31**

The synthetic procedure for compounds **29**, **30**, **31** in shown in Figure 5.

##### 4-(4-Nitrophenyl)morpholine (**29**)

An amount of 1.015 g (7.2 mmol) of 1-fluoro-4-nitrobenzene, 1 mL (11.6 mmol) of morpholine, and 3 mL (21.5 mmol) of triethylamine were dissolved in 7.5 mL of acetonitrile and refluxed at 80 °C for 3 h. The reaction was slowly cooled and poured into 30 mL of water and extracted twice with 30 mL of EtOAc. The organic layers were combined, washed with brine, dried over sodium sulphate, and concentrated. This resulted in a yellow solid (yield: 86%). ^1^H NMR (400 MHz, CDCl_3_) δ 8.19–8.10 (m, 2H), 6.88–6.79 (m, 2H), 3.90–3.83 (m, 4H), 3.41–3.34 (m, 4H).

##### 4-Morpholinoaniline (**30**)

An amount of 1.280 g (6.15 mmol) of **29** was dissolved in 20 mL of absolute ethanol under argon atmosphere. Pd/C 10 m/m% was quickly added to the mixture, and the reaction was stirred overnight under hydrogen. Once the reaction was complete, the Pd/C complex was filtered off, and the filtrate was concentrated to yield a purple solid (yield: 87%). ^1^H NMR (400 MHz, DMSO-*d*_6_) δ 6.70–6.64 (m, 2H), 6.52–6.45 (m, 2H), 4.56 (s, 2H), 3.70–3.65 (m, 4H), 2.89–2.82 (m, 4H).

##### 4-(4-Isothiocyanatophenyl)morpholine (**31**)

An amount of 489 mg (2.7 mmol) of **30** and 1 mL (7.2 mmol) of triethylamine were dissolved in 5 mL of THF under argon atmosphere. Then, 0.4 mL (6.6 mmol) of carbon disulfide was slowly added to the solution at 0 °C. The reaction was stirred overnight and monitored by TLC (EtOAc: petroleum ether 1:1). When **30** was completely converted into the corresponding dithiocarbamate salt, 695 mg (3.6 mmol) of tosyl chloride was added to the mixture and the reaction was stirred for 1 h. The mixture was poured into a 1 M HCl solution (10 mL) and extracted 3 times with 10 mL of MTBE. The organic layers were combined, dried, and concentrated. Then, tosyl chloride was removed from the mixture by column chromatography eluting with petroleum ether/EtOAc 0–20%. The product was concentrated and dried to yield a pale-yellow solid (yield: 70%). ^1^H NMR (400 MHz, DMSO-*d*_6_) δ 7.31–7.22 (m, 2H), 6.97–6.88 (m, 2H), 3.72–3.65 (m, 4H), 3.15–3.08 (m, 4H).

### 3.2. Molecular Modeling

#### 3.2.1. Compound Library

Drug-like small molecule libraries from Asinex (Winston Salem, NC, USA), ChemBridge (San Diego, CA, USA), Enamine (Riga, Latvia), Life chemicals (Niagara-on-the-Lake, ON, Canada), Key Organics (Camelford, UK), Maybridge (Thermo Fisher Scientific Inc., Waltham, MA, USA), Vitas-M (Causeway Bay, Hong Kong), and Pharmeks (Moscow, Russia), were downloaded in SDF format. These libraries were merged, and duplicates removed, which resulted in a library containing 2,081,456 compounds. For these compounds, a library of conformers was generated using OMEGA software (Release 2.5.1.4, OpenEye Scientific Software, Inc., Santa Fe, NM, USA; www.eyesopen.com) [35] using the default settings, which resulted in a maximum of 200 conformers per ligand.

#### 3.2.2. Structure-Based Virtual Screening

For docking with FRED software (Release 3.2.0.2, OpenEye Scientific Software, Inc., Santa Fe, NM, USA; www.eyesopen.com) [36], the UDP-GlcNAc binding site in OGT (PDB entry: 4N39) [37] was prepared using MAKE RECEPTOR (Release 3.2.0.2, OpenEye Scientific Software, Inc., Santa Fe, NM, USA; www.eyesopen.com). The grid box around the UDP bound in the OGT crystal structure was generated automatically and adjusted to contain also the GlcNAc binding site. This resulted in a box with the following dimensions: 22.00 Å × 16.33 Å × 24.33 Å and a volume of 8743 Å^3^. For “Cavity detection”, a slow and effective “Molecular” method was used for the detection of binding sites. The inner and outer contours of the grid box were also calculated automatically using the “Balanced” settings for the “Site Shape Potential” calculation. The inner contours were disabled. Ala896 was defined as the hydrogen bond donor and acceptor constraint for the docking calculations. The co-crystallized ligand, UDP, was docked to the prepared receptor using FRED (Release 3.2.0.2. OpenEye Scientific Software, Inc., Santa Fe, NM, USA) [36] with an RMSD of 1.45 Å, thus validating the docking protocol. The small molecule library, prepared by OMEGA, was then docked at the prepared UDP-GlcNAc-binding site of OGT (PDB entry: 4N39) [37] using FRED. The docking resolution was set to high, other settings were set as default. A hit list of the top 1000 ranked molecules was retrieved, and the best ranked FRED-calculated pose for each compound was inspected visually and used for analysis and representation.

#### 3.2.3. Docking

For docking with FRED software (OEDOCKING 3.3.1.2, OpenEye Scientific Software, Inc., Santa Fe, NM, USA; www.eyesopen.com) [29], the OGT-binding site (PDB entry: 4GYY) [31] was prepared using MAKE RECEPTOR (Release 3.3.1.2, OpenEye Scientific Software, Inc., Santa Fe, NM, USA; www.eyesopen.com) [29]. The grid box around the ligand UDP-5S-GlcNAc bound in the OGT crystal structure was generated automatically and was not adjusted. This resulted in a box with the following dimensions: 21.67 Å × 18.33 Å × 21.33 Å and a volume of 8474 Å^3^. For “Cavity detection”, a slow and effective “Molecular” method was used for the detection of binding sites. The inner and outer contours of the grid box were also calculated automatically using the “Balanced” settings for the “Site Shape Potential” calculation. The inner contours were disabled. Ala896 was defined as the hydrogen bond donor and acceptor constraint for the docking calculations. The ligands were prepared by OMEGA (Release 3.3.1.2, OpenEye Scientific Software, Inc., Santa Fe, NM, USA; www.eyesopen.com) and were then docked to one of the prepared binding sites of OGT using FRED (default settings). The resulting file was saved in SDF format and edited with PyMOL (The PyMOL Molecular Graphics System, Version 1.5.0.3 Schrödinger, LLC, New York, NY, USA).

#### 3.2.4. Molecular Dynamics Simulation

The MD simulation of the OGT in complex with Vs-51 was performed using the NAMD package (version 2.9 [38]) and the CHARMM22 force field [39,40]. Molecular mechanics parameters for Vs-51 were calculated using the ParamChem tool [41,42,43]. The system for the MD simulation was prepared using psfgen in VMD (version 1.9.1 [44]). The complex was first embedded in a box of TIP3P water molecules and then neutralized by the addition of NaCl. The MD simulation was run in the NPT ensemble using the periodic boundary conditions. Temperature (300 K) and pressure (1 atm) were controlled using the Langevin dynamics and Langevin piston methods, respectively. Short-range and long-range forces were calculated every 1 and 2 time-steps, respectively, with a time step of 2.0 ps. The smooth particle mesh Ewald method was used to calculate the electrostatic interactions [45]. The short-range interactions were cut off at 12 Å. The chemical bonds between hydrogen and the heavy atoms were held fixed using the SHAKE algorithm [46]. The simulation consisted of three consecutive steps: (i) solvent equilibration for 1 ns; (ii) complete system equilibration for 1 ns; and (iii) an unconstrained 100 ns production run. For structure-based pharmacophore modeling, 1000 frames from the production run were saved separately and used for interaction analysis.

#### 3.2.5. Structure-Based Pharmacophore Modeling

The 100 ns MD trajectory of OGT in complex with Vs-51 was used for pharmacophore feature analysis using LigandScout 4.4 Expert [47], which resulted in 1000 structure-based pharmacophore models.

### 3.3. cLogP Prediction

SwissADME (http://www.swissadme.ch/ accessed on 14 March 2022) webserver was used to calculate the cLogP values of compounds (consensus LogP_o/w_, average of the values obtained with five different prediction methods) [34].

### 3.4. Fluorescent Activity Assay

The fluorescent activity assay was performed as published [32]. OGT reactions were carried out in a 25 μL final volume containing 2.8 µM of glycosyl donor BFL-UDP-GlcNAc, 50–200 nM of purified full-length OGT, and 9.2 μM of glycosyl acceptor HCF-1 serine in OGT reaction buffer (1 × PBS pH 7.4, 1 mM DTT, 12.5 mM MgCl_2_). The reactions were incubated at room temperature for 1 h in the presence of different concentrations of inhibitors (the inhibitors were pre-incubated with OGT for at least 5 min). The reactions were then stopped by a mix of UDP at a final concentration of 2 mM and a solution of Nanolink^®^ magnetic streptavidin beads (Vector Laboratories, Burlingame, CA, USA) (2 µL of stock solution per reaction). After incubation at room temperature for 30 min, the beads were immobilized on a magnetic surface and washed thoroughly with PBS-tween 0.01%. Finally, the beads were resuspended in PBS-tween 0.01% and transferred into a microplate. Fluorescence was read at Ex/Em = 485/530 with a POLARstarR^®^ Omega microplate reader (BMG LABTECH, Ortenberg, Germany) or a Synergie H4 Hybrid Reader (BioTek, Winooski, VT, USA). The data were normalized and plotted with GraphPad Prism 8.2.1 software. The concentration of the inhibitor, where the residual activity of the enzyme is 50% (IC_50_), was calculated using a nonlinear regression-based fitting of inhibition curves using the (inhibitor) vs. response-variable slope (four parameters).

### 3.5. Cell-Based Assays

#### 3.5.1. Cell Cultures

AMO1 and K562 (ATCC, Manassas, VA, USA) cell lines were cultured in RPMI-1640 (Sigma-Aldrich, St. Louis, MO, USA), supplemented with 10% fetal bovine serum (Gibco, Life Technologies, Paisley, UK), 2 mM of L-glutamine, 100 U/mL of penicillin, 100 µg/mL of streptomycin (all Sigma-Aldrich, St. Louis, MO, USA) in a humidified chamber at 37 °C and 5% CO_2_.

#### 3.5.2. Metabolic Activity Assay

K562 and AMO1 were seeded into 96-well plates at a density of 8.000–10.000 and 10.000–12.000 cells per well, respectively, and treated with compounds of interest or corresponding vehicle as control. The metabolic activity was assessed after 72 h treatment using the CellTiter96 Aqueous One Solution Cell Proliferation Assay (Promega, Madison, WI, USA). The absorbance was measured at 492 nm on an automated microplate reader Synergy™ 4 Hybrid Microplate Reader (BioTek, Winooski, VT, USA). The data were normalized to the control sample and results were presented as the percentage of the metabolic activity (Mean ± SD) of two independent experiments, each conducted in duplicate.

#### 3.5.3. Statistical Analysis

Statistical analysis was performed with Student’s *t*-test between the untreated control vs. each treated sample, using GraphPad Prism 9. A *p* < 0.05 was considered statistically significant.

#### 3.5.4. SDS-PAGE

AMO1 was cultured at a density of 1 × 10^6^ cells per mL and treated with compounds of interest or corresponding vehicles. After 4 h, 2 × 10^6^ cells were harvested and centrifuged at 2400 rpm for 5 min. Afterward, the cells were resuspended in ice-cold PBS and centrifuged at 2400 rpm for 5 min. Cell pellets were lysed on ice using modified RIPA buffer, consisting of 50 mM of Tris–HCl, pH 8.0, 150 mM of NaCl, 1% NP-40, 0.5% Na-deoxycholate, 0.1% SDS, 1 mM of EDTA, 1 × Halt Phosphatase inhibitor cocktail, and 1 × Halt Protease inhibitor cocktail (Thermo Scientific, Waltham, MA, USA). Then, the lysates were sonicated, rocked on ice for 30 min, and centrifuged at 15000× *g* at 4 °C for 20 min. The samples containing 20 µg of protein were denaturated at 96 °C for 5 min in a sample loading buffer (3% SDS, 10% glycerol, 62.5 mM of Tris–HCl, pH 6.8, 5% 2-mercaptoethanol, 0.1% bromphenol blue) and loaded on 8% SDS-polyacrylamide gels. Electrophoresis was carried out in Tris-glycin buffer at 100 V, followed by a wet transfer to nitrocellulose membranes (GE Healthcare Life Science, Uppsala, Sweden). The SeeBlue^®^ Plus2 pre-stained reagent (Invitrogen, Waltham, MA, USA) was used to determine the molecular weights of separated proteins. Nonspecific binding sites were blocked for 1 h at room temperature in 3% bovine serum albumin (Sigma-Aldrich, St. Louis, MO, USA) in tTBS (TBS, 0.1% Tween; Sigma-Aldrich, St. Louis, MO, USA). The membranes were then washed and incubated overnight at 4 °C with gentle stirring in a solution containing appropriate primary antibodies. The next day, the membranes were washed three times with 0.1% Tween in TBS and incubated for 1 h at room temperature with the corresponding dilution of a secondary antibody conjugated to horseradish peroxidase (Cell Signaling Technology, Danvers, MA, USA) in a 5% solution of skim milk powder (Merck, Kenilworth, NJ, USA) (TBS, 0.1% Tween). After incubation, the membranes were washed 5-times in 0.1% Tween in TBS, and then the SuperSignal West Femto substrate (ThermoScientific, Waltham, MA, USA) was added. The chemiluminescent signal was acquired on the Uvitec Cambridge Alliance chemiluminometer (Uvitec, Lodi, NJ, USA). The band intensities were quantified using the Uvitec Imager. To ensure the equal loading of proteins, the membranes were stripped with a stripping buffer (100 mM 2-mercaptoethanol, 2% SDS, and 62.5 mM Tris/HCl, pH = 6.8) for 45 min at 50 °C and re-probed with antibodies as described above.

The antibodies and their dilutions used were as follows: anti-O-GlcNAcylation (CTD110.6; 1:800; BioLegend, San Diego, CA, USA), anti-ß-tubulin (2146; 1:1000; Cell Signaling Technology, Danvers, MA, USA), anti-mouse IgG-HRP (7076, 1:10,000) and anti-rabbit IgG-HRP (7074; 1:10,000).

## 4. Conclusions

In conclusion, we discovered a new series of OGT inhibitors through a comprehensive virtual screening campaign, followed by in vitro testing of selected virtual hits with the fluorescent activity assay. The most potent and synthetically versatile hit was selected as the basis to design a small series of derivatives with the aim of defining its structure-activity relationships. The selected hit (**1**) incorporates a novel uridine mimetic scaffold and has an IC_50_ value of 7 µM, making it an excellent starting point from which more potent OGT inhibitors could be obtained. This is significant, since the number of OGT inhibitors reported in the literature is limited, and most of them are structurally related to each other. The compound, however, appeared to show some off-target effects in AMO1 cells. Nevertheless, this series of compounds have a number of promising properties, including molecular weights in the 378–501 range and few hydrogen bond donors and acceptors. While the cLogP values are currently relatively high in the range of 2.56–3.76, this can likely be decreased by subsequent modifications, including, for example, hydrophilic groups, as seen for the phenol-containing compound **6**. Our preliminary SAR study provides valuable information about the binding mode of these compounds, which can be used to develop more potent and selective inhibitors in the future that should exhibit cellular activity.

## Data Availability

Publicly available datasets were analyzed in this study. This data can be found here: https://www.rcsb.org/structure/4n39 (accessed on 17 March 2022); https://www.rcsb.org/structure/4GYY (accessed on 17 March 2022).

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
