# Peer review of "Discovery of a New Drug-like Series of OGT Inhibitors by Virtual Screening"

_molecules, 2022, doi:10.3390/molecules27061996_

Round 1

Reviewer 1 Report

In this study, the authors discovered a series of OGT inhibitors through a virtual screening and in vitro testing with fluorescent activity assay, and then synthesized a small series of derivatives of Vs-51 with the aim to explore its structure-activity relationship. The research work is a relatively interesting and groundbreaking research paper on the whole. However, some results of this paper is not mighty enough.

  1. The diversity of substituents of the compounds in synthesized library is limited.
  2. The IC50 of the positive control, that is OSMI-4, did not be provided.
  3. The result of cell-based assay as shown in Fig. 4A looks like false positive. This figure lacks of negative controls, and OSMI-4 does not display metabolic inhibitory activity as positive control.
  4. Fig 3 does not be mentioned in the manuscript. 

Author Response

Point-by-point reply to Reviewer 1 comments and suggestions

(also in word file)

In this study, the authors discovered a series of OGT inhibitors through a virtual screening and in vitro testing with fluorescent activity assay, and then synthesized a small series of derivatives of Vs-51 with the aim to explore its structure-activity relationship. The research work is a relatively interesting and groundbreaking research paper on the whole. However, some results of this paper is not mighty enough.

  1. The diversity of substituents of the compounds in synthesized library is limited.

We agree with the reviewer that the compounds presented in the study represent a focused library. However, since our goal was to gain information and possibly optimize an already confirmed hit by varying two quite defined substitution sites, we believe that the number of substituents tested is appropriate, at least at this stage of the research. 

  1. The IC50 of the positive control, that is OSMI-4, did not be provided.

As a response to the reviewer's comment, we included the structure and IC50 value of the positive control OSMI-4a (free acid) and OSMI-4b (ester form) in Table 1.

  1. The result of cell-based assay as shown in Fig. 4A looks like false positive. This figure lacks of negative controls, and OSMI-4 does not display metabolic inhibitory activity as positive control.

In response to the reviewer's comment, we modified the caption of Figure 4a. We agree that OSMI-4b is not a positive control in this particular assay but rather a negative one to indicate that OGT inhibition does not, on its own, impair cellular metabolic activity in both assayed cell lines. Regarding the lack of negative controls in the picture, as we stated in the caption, the results are presented as the percentage of metabolic activity of the control cells treated with vehicle alone. The use of vehicle alone is the standard negative control for assays that measure cell viability like the MTS assay we use here. Finally, we don't believe that the data represents a false positive. Namely, the effect of compound 1 on metabolic activity was consistent in two different cell lines. Since this effect did not correlate with OGT inhibition according to western blot analysis, we concluded and noted in the manuscript that the MTS assay result with this compound is very likely a consequence of off-target effects.

  1. Fig 3 does not be mentioned in the manuscript.

As a response to the reviewer's comment, Figure 3 is now mentioned in the manuscript at line 189.

Reviewer 2 Report

  1. In this study authors have mentioned that about 2 million compounds from commercially available databases were used for virtual screening to find the potent molecule against OGT, but they also reported a set of compounds were synthesized and characterized. Which set of compounds have been used most prominently in this study?
  2. Why have authors not used the synthesized compounds in both in silico docking approaches?
  3. In vitro assay authors have mentioned generally, the desired compounds or standard compounds are used? What are the desired compounds that have been used in this assay evaluation?
  4. Are the top ranked compounds sharing the exact binding pocket of co-crystalized ligands of OGT?
  5. Provide information on the chemistry of both synthesized and screened compounds in more detail.
  6. Authors have mentioned that the cation-dipole interaction with the side chain of Lys898 could be beneficial; in which mechanism has it been beneficial to this interaction?
  7. The detailed abbreviations of some short forms are not provided in the manuscript and it makes reading the manuscript difficult.
  8. Authors recommended performing molecular simulation of top lead molecules to evaluate the stability. 
  9. Increase the figure of in vitro assays.

Author Response

Point-by-point reply to Reviewer 2 comments and suggestions

(also in word file)

  1. In this study authors have mentioned that about 2 million compounds from commercially available databases were used for virtual screening to find the potent molecule against OGT, but they also reported a set of compounds were synthesized and characterized. Which set of compounds have been used most prominently in this study?

The library of compounds described in Table 1 was built by selecting the most promising hits from the virtual screening of commercially available compounds within datasets. After virtual screening, the chosen compounds were purchased and screened in vitro as OGT inhibitors. As a follow up to that step, a focused library of analogues of the most potent hit (Vs-51) was synthesized and tested in vitro for OGT inhibition. The results of the IC50 measurements for these additional compounds are presented in Table 2. We have clarified this by improving our explanation of the steps we took and correcting their description in the introduction paragraph (lines 61-64). 

  1. Why have authors not used the synthesized compounds in both in silico docking approaches?

We are sorry if this was perhaps not stated clearly in the manuscript. The small library of synthesized compounds was indeed docked with OpenEye docking tools FRED and Hybrid. We have clarified this in the text in lines 118-119: “OpenEye docking tools FRED and Hybrid were used to guide us in designing a small new analogue library based on the hit Vs-51.”.

  1. In vitro assay authors have mentioned generally, the desired compounds or standard compounds are used? What are the desired compounds that have been used in this assay evaluation?

The known OGT inhibitors OSMI-4a (free acid) and OSMI-4b (ester form) were used as positive controls for the fluorescent activity assay. To make this information more evident, we integrated Table 1 with the structures of both its ester (OSMI-4b) and acidic form (OSMI-4a) and their corresponding IC50 values.

  1. Are the top ranked compounds sharing the exact binding pocket of co-crystalized ligands of OGT?

Yes, all the top ranked compounds are predicted to bind in the same pocket as the co-crystallized ligand of OGT. The virtual screening experiment was designed to discover novel chemotypes that could bind into this pocket, so the algorithm identified and ranked the compounds that are predicted to do so. We have clarified this point in the text at lines 77-82 and in the Figure 2, which clearly shows an overlap of compound 1 binding pose with co-crystalized UDP-5S-GlcNAc.

  1. Provide information on the chemistry of both synthesized and screened compounds in more detail.

We integrated the caption of Scheme 1 with additional information about the synthetic procedures. We also provided more details about the compounds we purchased from different vendors in a new table in the supplementary information (Table S2).

 6. Authors have mentioned that the cation-dipole interaction with the side chain of Lys898 could be beneficial; in which mechanism has it been beneficial to this interaction?

The introduction of chlorine in position 6 did not prove to be beneficial for binding. Since the formation of a cation-dipole interaction was more a hypothesis than a prediction, we reformulated our initial statement (lines 146-149).

  1. The detailed abbreviations of some short forms are not provided in the manuscript and it makes reading the manuscript difficult.

The list of abbreviations at the end of the manuscript was expanded with additional terms.  

  1. Authors recommended performing molecular simulation of top lead molecules to evaluate the stability.

In response to the reviewer's comment, we note that we performed a molecular dynamics simulation for the most potent hit (Vs-51). The results are reported in the manuscript and we draw attention to these data more clearly in lines 125-142.

  1. Increase the figure of in vitro assays.

The size of Figure 4 was increased. 

Reviewer 3 Report

Authors presented a paper  based on a very simple approach (only docking) for selecting some compounds showed weak activity against the target.

-intro a figure with the structure of known ligands should be reported

-The numbering of compounds should be rigorously checked. In Figure 2 "Representation of 6 (purple)" compound 6 was not introduced. I suggest to enclose the original name (Vs-X) and the correspondent number in the same table since in this current form it is very hard.

-there is no a control (positive/negative in the test presented in Table 2

-number of independent experiments should be reported for all the tests performed

-selection criteria for selecting docking hits should be reported and how the authors selected the threshold.

-toxicity of best performing compounds should be evaluated in healthy cells.

-it is quite strange that compound purchased showed 10-fold decrease of activity. The hypothesis about different purity should be confirmed reporting a purity of all purchased compounds (provided by vendors) and the synthesized ones.

-methods - authors reported the calculation pf clogp, it is not clear how this parameter helped authors in selecting compounds.

-docking protocol was not validates. Usually redocking using known crystallized ligands should be performed in order to assess docking performances.

-English grammar should be checked

in general authors reported a screening procedure based only on docking finding some compounds with weak activity against the target. Authors should discuss the difference between the selected hits and known ligands and how this work can be useful in the field

Author Response

Point-by-point reply to Reviewer 3 comments and suggestions

(also in word file)

Comments and Suggestions for Authors

Authors presented a paper based on a very simple approach (only docking) for selecting some compounds showed weak activity against the target.

  1. Intro a figure with the structure of known ligands should be reported.

We have introduced a summary of the known OGT ligands, including their structures, in the Supplementary information (Table S1). We did not include the table in the main text, as it would enlarge the introduction beyond the desired size and scope.

  1. The numbering of compounds should be rigorously checked. In Figure 2 "Representation of 6 (purple)" compound 6 was not introduced. I suggest to enclose the original name (Vs-X) and the correspondent number in the same table since in this current form it is very hard.

We have checked the numbering of compounds and the original name of compound 1 (Vs-51) was specified in Table 2 as suggested by the reviewer.

  1. There is no a control (positive/negative in the test presented in Table 2.

The control used in the assay presented in Table 2 is now stated in the table's caption.

  1. Number of independent experiments should be reported for all the tests performed.

We have verified that the number of independent experiments is reported for all the tests performed.

  1. Selection criteria for selecting docking hits should be reported and how the authors selected the threshold.

As described in the Results and Discussion section of the manuscript, we first ranked the virtual screening hits based on the FRED scoring function and then selected the top 120 hits. This reduction allowed us to carefully inspect the predicted binding poses of a manageable number of virtual hits. In response to the reviewer's comment, this is now clarified in the discussion (lines 85-87). Subsequently, the molecules were clustered into 9 different chemotype families based on their uridine mimetic scaffold, and one to three hits were purchased from each family based on their synthetic accessibility and chemical diversity.

  1. Toxicity of best performing compounds should be evaluated in healthy cells.

We are mindful of the importance of evaluating the toxicity in healthy cells of novel biologically active compounds. However, at this stage it was important to elucidate whether our best hit displays off-target effects in two human cell lines. For this reason, we concluded that the optimization of the potency and selectivity of the new OGT inhibitor should be a priority. Should this lead to the successful development of a potent and selective inhibitor into lead phase of the development, we agree that it would be crucial to assess its toxicity in healthy cells.

  1. It is quite strange that compound purchased showed 10-fold decrease of activity. The hypothesis about different purity should be confirmed reporting a purity of all purchased compounds (provided by vendors) and the synthesized ones.

We were also surprised to observe such a difference in the activity of the purchased and the resynthesized compound. Although the purity declared by the vendor was approximately 90%, the actual value obviously deviates, as the purity was probably measured by the vendor right after the synthesis, not prior to purchase order. This is the reason why we choose to resynthesize and test again the hit we selected for the optimization. An alternative, although less likely, explanation could be the instability of the molecule in DMSO solution. As recommended by the reviewer, we included the information provided by the vendors in the supporting information (Table S2).

  1. Methods - authors reported the calculation pf clogp, it is not clear how this parameter helped authors in selecting compounds.

cLogP values were not used to select the compounds, but they were calculated to help us evaluate the possible improvement areas for the hit expansion process.

  1. Docking protocol was not validates. Usually redocking using known crystallized ligands should be performed in order to assess docking performances.

We have validated the docking protocol before performing the virtual screening, but did not report the details. As a response to the reviewer's comment, the docking protocol validation is now reported in the materials and methods section (lines 590-593).

  1. English grammar should be checked

The manuscript was proofread again by some of the authors who are native English speakers. Some minor grammar errors were corrected, which improved the paper.

  1. In general authors reported a screening procedure based only on docking finding some compounds with weak activity against the target. Authors should discuss the difference between the selected hits and known ligands and how this work can be useful in the field

The number of OGT inhibitors reported in the literature is limited, and most of them are structurally related to each other. Therefore, it is crucial to identify new chemotypes that can bind to the enzyme's active site. The compounds presented in this work display novel uridine mimetic scaffolds that can be used as the basis to design more potent and selective OGT inhibitors. Furthermore, with an IC50 value of 7 µM, compound 1 is amongst the most promising OGT inhibitors reported to date, as it is only an order of magnitude weaker than OSMI-4a in the same assay (lines 167-169). We have provided more discussion of this point in the manuscript (lines 238-240).

Round 2

Reviewer 1 Report

 The compound code "OSMI-4" in figure 4 should be revised as "OSMI-4b" as caption described.

Author Response

We are thankful for the positive review from the Reviewer 1. The compound code "OSMI-4 " in Figure 4 was changed to "OSMI-4b" as recommended by the reviewer. 

Reviewer 2 Report

Can be accepted in present form. I am happy to endorse this revised manuscript.!

Best wishes for the authors

Author Response

We are thankful for the positive review from the Reviewer 2.

Reviewer 3 Report

the manuscript was improved but some points remain to be addressed:

-Authors performed MD simulation, reporting RMSD only for the ligand. Authors should provide RMSD also for the protein

-the presentation of the results is still confusing, authors reported in the docking picture (Figure 2) compound 6 that is a compound obtained in the second step. while a single compound (9) is reported in another figure. I suggest to better organize the discussion enclosing the first structure obtained in the first figure and the other figure of optimized compounds in another picture. In this way is very confused.

Author Response

  1. In response to the reviewer's comment, we reported the protein RMSD values in Figure S3.

  1. According to the reviewer's suggestion, the docking picture of compound 6 was moved from Figure 2c to Figure 3a.